# Physicochemical and Functional Properties of Thermal-Induced Polymerized Goat Milk Whey Protein

**DOI:** 10.3390/foods12193626

**Published:** 2023-09-29

**Authors:** Mu Tian, Xiaomeng Sun, Jianjun Cheng, Mingruo Guo

**Affiliations:** 1College of Food Science and Technology, Southwest Minzu University, Chengdu 610041, China; tm18646175069@163.com; 2Key Laboratory of Dairy Science, Northeast Agricultural University, Harbin 150030, China; sunxm@neau.edu.cn (X.S.); cheng577@163.com (J.C.); 3Department of Nutrition and Food Sciences, College of Agriculture and Life Sciences, University of Vermont, Burlington, VT 05405, USA

**Keywords:** goat milk whey protein, heat treatment, polymerization, physicochemical property, functional property

## Abstract

Goat milk whey protein products are a hard-to-source commodity. Whey protein concentrate was directly prepared from fresh goat milk. The effects of the heating temperature (69–78 °C), time (15–30 min), and pH (7.5–7.9) on the physicochemical and functional properties of the goat milk whey protein were investigated. The results showed that the particle size of the samples significantly increased (*p <* 0.05) after heat treatment. The zeta potential of polymerized goat milk whey protein (PGWP) was lower than that of native goat milk whey protein. The content of the free sulfhydryl groups of PGWP decreased with increasing heating temperature and time, while an increase in surface hydrophobicity and apparent viscosity of PGWP were observed after heat treatment. Fourier Transform Infrared Spectroscopy analysis indicated that heat treatment and pH had considerable impacts on the secondary structure of goat milk whey protein. Transmission electron microscope images revealed that heat induced the formation of a large and uniform protein network. Additionally, the changes in the physicochemical and structural properties contributed to the improvement of the emulsifying and foaming properties of goat milk whey protein after heat treatment. The results may provide a theoretical basis for the applications of polymerized goat milk whey protein in related products.

## 1. Introduction

Goat milk whey protein is a valuable component of goat milk, and has gained increasing attention due to its unique nutritional and functional properties [1]. As a rich source of essential amino acids and bioactive peptides, goat milk whey protein products have been used in the food, pharmaceutical, and nutraceutical industries [2]. The use of microfiltration (MF) and ultrafiltration (UF) techniques enables the extraction of native goat milk whey proteins from raw goat milk [3], offering advantages such as energy efficiency, mild processing conditions, and minimized environmental contaminants [4]. Extensive research has shown that membrane separation technology yields goat milk whey protein with superior nutritional and functional properties [5].

The functional properties of native whey protein are inherently limited by its compact structure [6]. Heat treatment is widely employed in dairy processing as a common method for improving protein functionality. Among these heat-induced modifications, thermal-induced polymerization stands out as a process involving controlled heating [7,8], leading to the formation of larger protein aggregates. This polymerization process significantly impacts the physicochemical and functional properties of whey protein, including solubility, gelation, emulsification, and foaming properties [9]. Exploring the effects of the thermal-induced polymerization of goat milk whey protein holds the potential to provide valuable insights into its diverse applications and enhance its functionality in various food and beverage formulations. Our previous research demonstrated the potential of polymerized goat milk whey protein (PGWP) as an effective thickening agent, enhancing the viscosity and minimizing syneresis in yogurt [10]. Furthermore, PGWP showed promise as an ideal carrier material for delivering bioactive substances such as soy isoflavones [11], thereby extending its utility in functional food applications. Additionally, the controlled thermal denaturation of whey protein can serve as an efficient fat substitute and stabilizer in yogurt formulations [12]. It is worth noting that Caner’s work revealed that whey protein isolate coatings can act as a protective barrier, prolonging the shelf life of eggs [13].

While previous studies have predominantly focused on investigating the physicochemical and functional properties of cow milk whey protein [14,15], data on the functionality of goat milk whey protein are very limited. Considering the compositional and structural differences between goat and cow milk whey proteins, it is necessary to investigate the functional properties of goat milk whey proteins and their thermal-induced polymerization behaviors.

The functional properties of goat milk whey protein are closely intertwined with its physicochemical attributes and can be further enhanced through various modifications. Therefore, the objectives of this study were to investigate the physicochemical and functional properties of thermal-induced polymerized goat milk whey protein.

## 2. Materials and Methods

### 2.1. Materials

Raw goat milk (≥8.15% nonfat solids, 3.86% protein, and 4.02% fat, *w*/*v*) was purchased from a local farm (Feihe Dairy Industry Co. Ltd., Harbin, China).

### 2.2. Preparation of Goat Milk Whey Protein Concentrate

Raw goat milk was subjected to heat treatment at 55 °C and subsequently separated into skimmed goat milk and cream using a separator (SA 10-T, Frautech SRL, Thiene, Italy). The skimmed goat milk was then subjected to microfiltration (MF) at 50 °C, employing a 0.1 μm filter. The MF permeate was further processed through ultrafiltration (UF) using a spiral-wound membrane with a 10 kDa molecular weight cut-off, resulting in a tenfold concentration [10]. To reduce the salt content, the UF retentate underwent electrodialysis (ED), successfully removing 85% of the salt. Finally, the concentrated goat milk whey protein was freeze-dried using a freeze dryer (Alpha 1–2, Marin Christ Inc., Osterode, Germany), yielding goat milk whey protein powder with the following composition: 80.99% protein, 18.67% lactose, and 0.34% ash (*w*/*w*).

### 2.3. Preparation of Polymerized Goat Milk Whey Protein (PGWP)

The goat milk whey protein powder was dissolved in deionized water with continuous stirring for 2 h at room temperature, resulting in a 10% (*w*/*v*) whey protein solution. This solution was then stored at 4 °C for 12 h to ensure complete hydration, and was then subsequently brought back to room temperature before use [11]. Three variables were studied. Variable (1) heating temperatures: 10% (*w*/*v*) whey protein solutions were adjusted to pH 7.7 and heated in a water bath to temperatures ranging from 69 to 78 °C for a duration of 20 min. Variable (2) heating times: the 10% (*w*/*v*) whey protein solutions were adjusted to pH 7.7 and heated to 75 °C in a water bath for durations ranging from 15 to 30 min. Variable (3) pH values: the 10% (*w*/*v*) whey protein solutions were adjusted to pH levels of 7.5, 7.7, and 7.9 using 1M sodium hydroxide and then heated to 75 °C for 25 min. Control protein solution samples were prepared following the same procedures mentioned above but without undergoing any heat treatment.

### 2.4. Particle Size and Zeta Potential Analysis

The particle size and zeta potential of the whey protein samples were carried out using a Malvern Zetasizer Nano ZS90 (Malvern Instruments Ltd., Worcestershire, UK) as described by Gélébarta et al. [16]. Sample solutions were diluted to 0.1% (*w*/*v*) with deionized water and stored at 25 °C for 30 min to ensure equilibrium. The refractive indexes for protein and water were 1.450 and 1.333, respectively. All measurements were performed in triplicate.

### 2.5. Determination of the Free Sulfhydryl Group

The amount of the free sulfhydryl group in the whey protein samples was measured as described by Ellman [17] with some modifications. Sample solutions were diluted to 1% (*w*/*v*) using demineralized water. A sample of 0.5 mL was mixed with 5 mL of urea buffer (8 M) and 20 μL of Ellman’s reagent. The mixtures were stored at room temperature for 20 min, and the absorbance was measured at 412 nm using a UV-Vis spectrophotometer (TU-1800, Beijing, China). The amount of the free sulfhydryl group was calculated as follows:μmol SH/g = (73.53 × A_412_ × D)/C(1)
where: A_412_ = absorbance value at 412 nm, D = dilution factor, C = protein concentration (mg/mL), 73.53 is obtained from 10^6^/(1.36 × 10^4^), 1.36 × 10^4^ = molar extinction coefficient.

### 2.6. Measurement of Surface Hydrophobicity

The surface hydrophobicity of the whey protein samples was determined using an 8-anilino-1-naphthalenesulfonic acid (ANS) probe following the method described by Haskard et al. [18]. The samples were diluted to concentrations ranging from 0.0125 to 0.1 g/kg. Fluorescence intensity measurements were performed at excitation and emission wavelengths of 365 nm and 484 nm, respectively, using a Spectrofluorometer (F-7000, Hitachi Ltd., Tokyo, Japan). The slope of the linear regression of fluorescence intensity versus protein concentrations was calculated and applied to the index of protein surface hydrophobicity (H0).

### 2.7. Rheological Properties Measurement

The rheological properties of the whey protein samples were assessed using a rheometer (Thermo Rheometer, San Jose, CA, USA) equipped with a 35-mm diameter plate at a controlled temperature of 25 °C. The flow tests for the samples involved measuring shear rates ranging from 0.1 to 1000 s^−1^. A time sweep test was conducted by maintaining a shear rate at 200 s^−1^ for 2 min according to Havea et al. [19] with some modifications.

### 2.8. Fourier Transform Infrared (FT-IR) Spectroscopy

The FTIR spectra of the whey protein samples were analyzed using a Nicolet 6700 FTIR spectrometer equipped with an attenuated total reflectance (ATR) ZnSe crystal (Thermo Electron Scientific Instruments Corporation, San Jose, CA, USA). Freeze-dried samples were ground into a fine powder. Measurements were recorded between 4000 and 400 cm^−1^, and the spectra were obtained at an average of 32 scans at a resolution of 4 cm^−1^ [20]. The spectral region ranging from 1600 to 1700 cm^−1^ was utilized to determine the secondary structure of the protein employing Peak FIT software. Within this range, bands located between 1610 and 1637 cm^−1^ and 1680–1692 cm^−1^ were assigned to β-sheet structures, while bands within the 1638–1648 cm^−1^ range were associated with random coil conformations. Additionally, bands spanning from 1649–1660 cm^−1^ were indicative of α-helix structures, and those in the 1660–1680 cm^−1^ range were attributed to β-turn structures. The area of each band was determined using a Gaussian function [21].

### 2.9. Transmission Electron Microscopy (TEM) Analysis

The microstructure of the whey protein samples was analyzed using a transmission electron microscope (H-7650, Hitachi High-Technologies, Tokyo, Japan) following a methodology based on Krebs [22] with some modifications. Samples were diluted to an appropriate concentration, and copper grids were placed in the diluted samples for 2 min and dyed with phosphotungstic acid for 30 s. Finally, the copper grids were examined using the transmission electron microscope.

### 2.10. Determination of the Emulsifying Properties of Goat Milk Whey Protein

The emulsifying activity index (EAI) and emulsifying stability index (ESI) of the whey protein samples were measured according to the method of Pearce and Kinsella [23] with some modifications. The 30 mL of protein solution (0.2%, *w*/*v*) was mixed with 10 mL of soybean oil. The mixture was then subjected to homogenization at a speed of 10,000 rpm for 1 min with a high-speed homogenizer (IKA, Labortechnik, Staufen, Germany).

A portion of 50 μL of the emulsion was pipetted from the bottom of the container at 0 and 10 min, respectively, and mixed with 5 mL 0.1% SDS. The absorbance of the emulsion was measured at 500 nm, using 0.1% SDS as a blank.
EAI (m^2^/g) = (2 × 2.303 × A_0_ × N)/(10,000 × θ × L × C)(2)
ESI (min) = (A_0_ × 10)/(A_0_ – A_10_)(3)
where N is the dilution factor of protein, θ is the volume fraction of the oil phase in the emulsion, L is the diameter of the colorimetric cup (cm), and A_0_, A_10_ are the absorbance of the emulsion at 0 and 10 min, respectively.

### 2.11. Determination of the Foaming Properties of Goat Milk Whey Protein

The foaming capacity (FC) and foam stability (FS) of the whey protein samples were determined using the method described by Ahmadi et al. [24] with some modifications. A volume of 20 mL of protein solution (5%, *w*/*v*) was placed in a 100-mL measuring cylinder and homogenized at 10,000 rpm for 2 min with a high-speed homogenizer (IKA, Labortechnik, Staufen, Germany).
FC (%) = [(V_0_ – 20)/V_0_] × 100 (4)
FS (%) = [(V_30_ – 20)/(V_0_ – 20)] × 100(5)
where V_0_ is the total volume directly after homogenization, V_30_ is the total volume after 30 min.

### 2.12. Statistical Analysis

Data were statistically analyzed and presented as mean ± standard deviations. An analysis of variance (*p <* 0.05) and a Tukey’s test were carried out using SPSS 20 software (SPSS Inc., Chicago, IL, USA). All figures were drawn by Origin 2020 (OriginLab Corp., Northampton, MA, USA).

## 3. Results and Discussion

### 3.1. Particle Size and Zeta Potential

The particle size and zeta potential of samples are shown in Figure 1. The particle size significantly increased (*p <* 0.05) with increasing heating temperature from 69 to 78 °C (Figure 1A). This increase may be attributed to the cross-linking and aggregation of denatured protein molecules following heat treatment. Croguennec et al. [25] reported that heating the protein solution at temperatures above 60 °C resulted in the unfolding of the protein secondary structure and exposed free sulfhydryl groups; both inter- and intra-molecular disulfide bonds may be formed from free sulfhydryl groups, which can promote the formation of larger aggregates. A gel network was observed when heat treatment was conducted at 78 °C after storage at 4 °C.

The particle size was found to increase when the heating time ranged from 15 to 30 min (Figure 1B). This result aligns with the findings using cow whey protein by Nicolai et al. [26], who observed an increase in the protein aggregate size with prolonged heating. Figure 1C demonstrates that there were no significant changes (*p >* 0.05) in the particle size of goat milk whey protein at different pH values. However, the particle size significantly decreased (*p <* 0.05) with increasing pH values after heat treatment. The reduction in particle size at higher pH values may be attributed to reduced protein attractive interactions as a result of increased high surface charge. Schmitt et al. [27] reported that bovine milk whey protein, heat-treated at 85 °C for 15 min, resulted in smaller-sized soluble aggregates with increasing pH.

Figure 1D,E demonstrates that the heated samples exhibited significantly higher absolute zeta potential values compared to native goat milk whey protein (control) (*p <* 0.05). The observed increase in absolute zeta potential can be attributed to the exposure of buried charged residues due to protein denaturation during heating [28]. However, variations in heating temperature and time did not yield a significant effect on the zeta potential of the PGWP samples (*p >* 0.05). This suggested that the changes in whey protein conformation induced by temperature and time may not be substantial enough to result in significant alterations in the zeta potential of the systems. Furthermore, after heat treatment, the zeta potential values of all samples were approximately −30 mV, indicating the relative stability of the heated samples [29].

In Figure 1F, the zeta potential of the goat milk whey protein was not significantly influenced by different pH values (*p >* 0.05). However, the zeta potential of the heated samples decreased (*p <* 0.05) as the pH varied from 7.5 to 7.9. This decrease in the zeta potential of the heated samples may be due to the increasing overall negative charge with an increase in pH values [30].

### 3.2. Free Sulfhydryl Group Content

Figure 2 illustrates the free sulfhydryl group (-SH) contents of the native and heated goat milk whey protein samples. The free sulfhydryl group contents decreased with increasing heating temperature (69 to 78 °C) and heating time (15 to 30 min) (Figure 2A,B), which resulted in the formation of disulfide bonds. The decrease in the free sulfhydryl group contents upon heat treatment was attributed to the formation of intermolecular and intramolecular disulfide bonds, resulting in a reduction in sulfhydryl group contents [31]. Regarded as a function of pH values, both the native and heated samples showed a decrease in free sulfhydryl group content as the pH values increased from 7.5 to 7.9 (Figure 2C). Mishyna et al. [32] showed that the total -SH groups decreased with increasing pH from 7 to 9 for whey protein isolates after heat treatment, which supports our results.

### 3.3. Surface Hydrophobicity

The surface hydrophobicity of the heated samples increased with higher heating temperatures ranging from 69 to 78 °C (Figure 3A). Additionally, the surface hydrophobicity of the heated samples significantly increased (*p* < 0.05) with prolonged heating times from 15 to 30 min (Figure 3B). This increase in surface hydrophobicity can be attributed to the altering of the conformation of the protein after heat treatment and the unfolding of the protein structure, resulting in the exposure of more hydrophobic groups [33].

Figure 3C shows that both the unheated and heated samples exhibited an increase in surface hydrophobicity as the pH values increased from 7.5 to 7.9 (*p* < 0.05), indicating that different pH values altered the characteristics of hydrophobic sites in the protein. Ahmad and Singh [34] suggested that an increase in pH values resulted in an elevation in surface hydrophobicity, which may be due to variations in the protein surface charge.

### 3.4. Apparent Viscosity

Figure 4 presents the apparent viscosity of the samples. The apparent viscosity of the heated samples increased with higher heating temperatures ranging from 69 to 78 °C (Figure 4A) and prolonged heating times from 15 to 30 min (Figure 4B), which is logical and consistent with the observed changes in particle size (Figure 1A,B). The process of heating leads to the disruption of various intramolecular bonds that stabilize the native goat milk whey protein structure. Beyond a certain temperature, protein molecules unfold and subsequently aggregate. These aggregates tend to be larger in size, exhibit greater asymmetry in shape, and possess a larger effective volume fraction compared to the native molecules, resulting in an increase in viscosity. However, it is essential to note that viscosity is influenced by numerous molecular properties, including size, shape, and flexibility [35]. All of these factors may be attributed to the observed increase in viscosity. Similar findings of increased apparent viscosity after heat treatment were reported by O’Loughlin et al. [36] on cow’s milk whey protein. In Figure 4C, the apparent viscosity of the heated samples decreased as the pH values increased from 7.5 to 7.9. This decrease can be attributed to the strong electrostatic repulsion at higher pH values [37].

### 3.5. Fourier-Transform Infrared (FTIR) Spectroscopy

The FTIR spectra of the heated samples exhibited similarities to those of the unheated samples (Figure 5). The amide I band, which falls within the spectral region between 1600 and 1700 cm^−1^, is sensitive to alterations in the secondary structure of proteins. It represents the stretching vibrations of C=O bonds and is influenced by hydrogen bonding interactions [38]. In the FTIR spectra of the unheated samples, a distinctive peak at 1633.96 cm^−1^ was prominently observed. Following heat treatment, significant blueshifts were observed as the heating temperature and time increased. These alterations in heat treatment conditions influenced the conformation of proteins by disrupting intramolecular hydrogen bonds, leading to a shift of the spectral peaks towards higher wavenumbers [39]. In addition, after heat treatment at 75 °C for 25 min, the absorption peaks of goat milk whey protein at pH 7.5, 7.7, and 7.9 were observed at 1635.41, 1637.34, and 1637.37 cm^−1^, respectively.

The secondary structures of goat milk whey protein encompass α-helix, β-sheet, β-turn, and random coil structures. Heat treatment was found to decrease the content of α-helix and increase the content of β-sheet in the goat milk whey protein compared to the unheated samples (Figure 6). The decrease in the α-helix structure after heating can be attributed to the rupture or weakening of hydrogen bonds within the α-helix structure, along with the formation of stronger intermolecular hydrogen bonds [38]. The increase in the β-sheet structure can be attributed to the aggregation of denatured protein molecules at higher temperatures [40]. Boye et al. [41] observed that heating promoted the unfolding of interior β-LG, leading to a decrease in the α-helix structure and enhanced formation of the intermolecular β-sheet structure.

When the goat milk whey protein was subjected to heat treatment at 75 °C for 25 min and adjusted to different pH values, the content of the α-helix structure decreased, while the β-sheet content increased. These results indicated that heat treatment and pH variations can induce changes in the secondary structures of goat milk whey protein [38].

### 3.6. Microstructure

The microstructure of the samples is shown in Figure 7. The native goat milk whey protein (control) displayed an irregular shape with slight aggregation. With an increase in heating time from 15 to 25 min, the particle size of the heated samples increased, and a uniform protein network was observed. However, when the heating time exceeded 25 min, the protein’s spherical structure transformed into a chain structure, resulting in the formation of a network of larger aggregates. Furthermore, increasing the heating temperature from 69 to 75 °C resulted in the formation of larger and more uniform protein particles. When the heat temperature increased to 78 °C, the particles agglomerated into large clusters. This phenomenon can be attributed to the protein interactions at higher temperatures, resulting in the agglomeration of protein particles [9]. As the pH values increased from 7.5 to 7.9, while maintaining the heat treatment at 75 °C for 25 min, the particle size of the aggregates significantly decreased. This trend of change was confirmed by dynamic light scattering measurements.

### 3.7. Emulsifying Activity Index (EAI) and Emulsion Stability Index (ESI)

The emulsifying activity index (EAI) reflects the protein’s ability to absorb at the oil-water interface, while the emulsion stability index (ESI) indicates the protein’s ability to remain at the oil-water interface after the emulsion is heated [31]. As shown in Figure 8, the EAI values of the heated samples were significantly lower than those of the unheated samples (*p <* 0.05). The decrease in EAI can be attributed to the formation of a thicker layer of proteins at the oil-water interface. Additionally, the EAI values significantly increased (*p <* 0.05) with increasing heating temperatures from 69 to 75 °C, reaching a maximum of 75 °C (Figure 8A). This suggests that more protein molecules could move to the oil-water interface, thereby enhancing the emulsifying ability.

On the other hand, increasing the heating time from 15 to 30 min at 75 °C had a negative impact on EAI values (Figure 8B). This can be attributed to the formation of large protein aggregates during heat treatments, which are unable to efficiently absorb the fat droplets [42]. Furthermore, a significant difference in EAI was observed between unheated and heated samples with varying pH values from 7.5 to 7.9 (*p <* 0.05) (Figure 8C). This difference may be due to the influence of different pH values on the surface charge of protein molecules, thereby affecting protein denaturation and aggregation.

According to the data presented in Figure 8, the ESI of the heated samples was significantly higher than that of the unheated samples (*p <* 0.05), with similar results also observed in bovine milk whey proteins. Jiang et al. [31] reported an increase in the ESI of bovine milk whey protein after heat treatment. The ESI values increased with higher heating temperatures (from 69 to 78 °C) (Figure 8D) and longer durations of heating (from 15 to 30 min) (Figure 8E). These results indicated that heat treatments can enhance the emulsion stability of goat milk whey protein. Moreover, the ESI values of both unheated and heated samples were significantly influenced by different pH values (*p <* 0.05) (Figure 8F).

### 3.8. Foaming Capacity (FC) and Foam Stability (FS)

The foaming capacity (FC) and foam stability (FS) data presented in Figure 9 demonstrate that the heated samples had significantly higher FC and FS values compared to the unheated samples (*p <* 0.05). The data indicated that heat treatments enhance the FC and FS of the goat milk whey protein. Furthermore, as the heating temperatures increased from 69 to 75 °C and the heating times extended from 15 to 25 min, both FC and FS showed significant increases (*p <* 0.05). The results suggested that the denaturation of protein molecules during heat treatment exposed more hydrophobic groups, leading to enhanced hydrophobic interactions and improved foam formation [43].

However, at a heating temperature of 78 °C or a heating time of 30 min, FC and FS decreased. This was attributed to the formation of aggregates reducing the availability of proteins to stabilize the gas-liquid interface at higher temperatures or during longer durations of heating [44]. Tosi et al. [45] discovered that heat treatment could enhance foam stability. However, it is important to note that the temperature should not exceed 85 °C to prevent excessive denaturation, which can subsequently diminish the foaming properties of whey. Additionally, different pH values had a significant impact on the FC and FS of the samples, which may be related to the changes in protein ionization, and the adsorption at the gas-liquid interface [46].

## 4. Conclusions

Goat milk whey protein concentrate was directly prepared from fresh milk using membrane separation technology. Heating treatments had major impacts on physicochemical properties, resulting in changes in the functional properties of the protein. The findings of this study indicated that thermally-induced polymerized goat milk whey protein may hold promise as a thickening agent and microencapsulating wall material in fermented product formulations.

## Figures and Tables

**Figure 1 foods-12-03626-f001:**
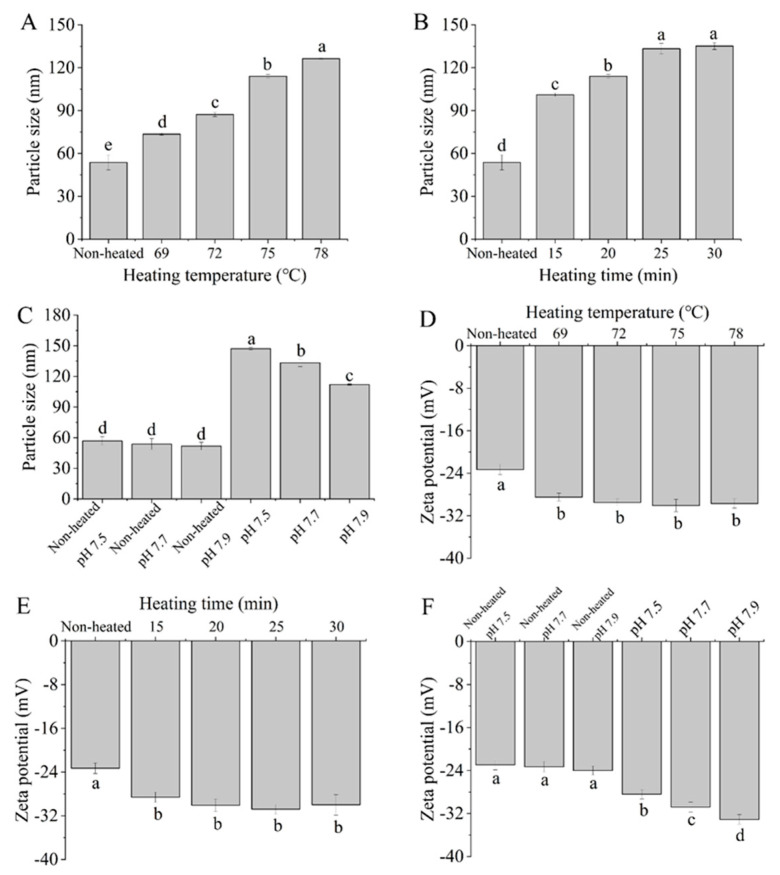
Heat−induced changes in the particle size and zeta potential of goat milk whey protein ((**A**,**D**) heating temperature, (**B**,**E**) heating time, (**C**,**F**) pH). Different letters are significantly different (*p <* 0.05). Error bars represent the standard deviation of the means.

**Figure 2 foods-12-03626-f002:**
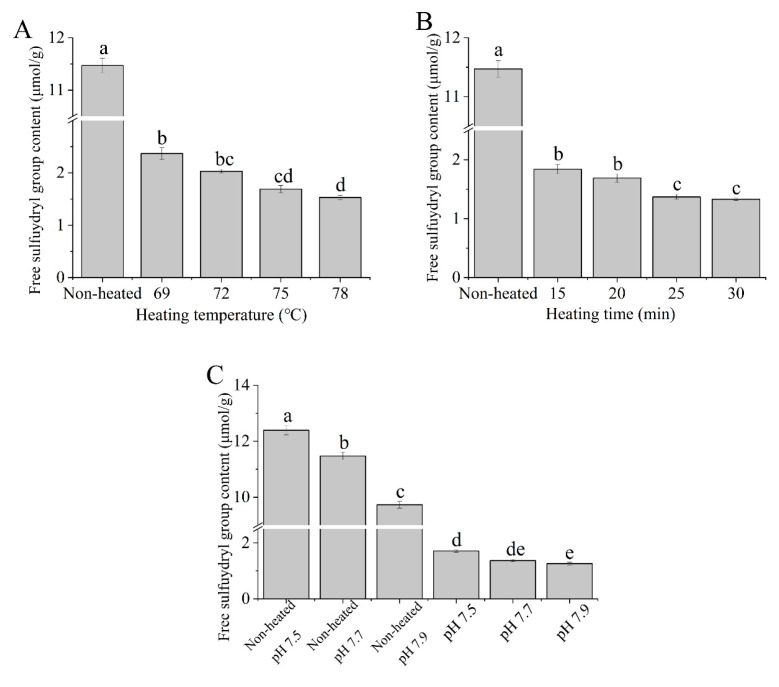
Heat−induced changes in the free sulfhydryl group contents of goat milk whey protein ((**A**) heating temperature, (**B**) heating time, (**C**) pH). Different letters are significantly different (*p <* 0.05). Error bars represent the standard deviation of the means.

**Figure 3 foods-12-03626-f003:**
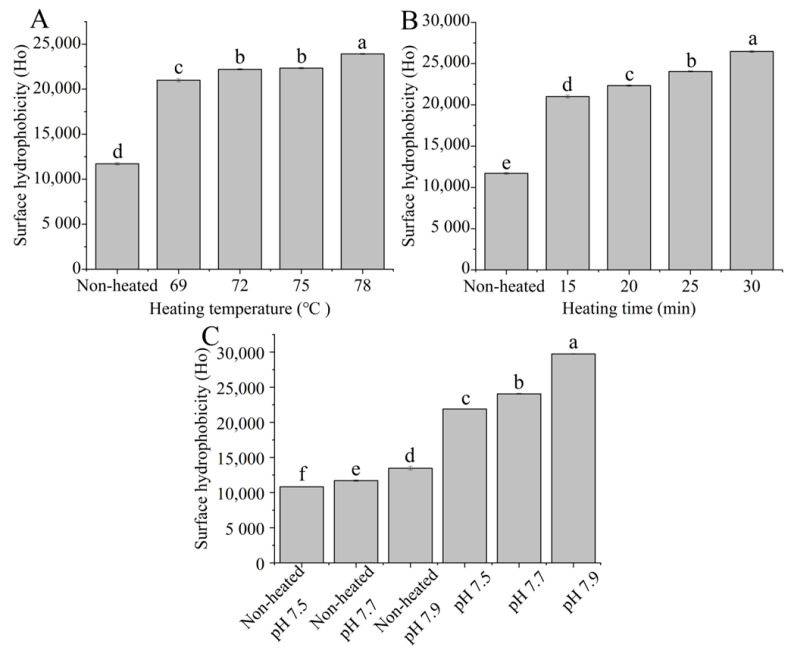
Heat−induced changes in the surface hydrophobicity of goat milk whey protein ((**A**) heating temperature, (**B**) heating time, (**C**) pH). Different letters are significantly different (*p <* 0.05). Error bars represent the standard deviation of the means.

**Figure 4 foods-12-03626-f004:**
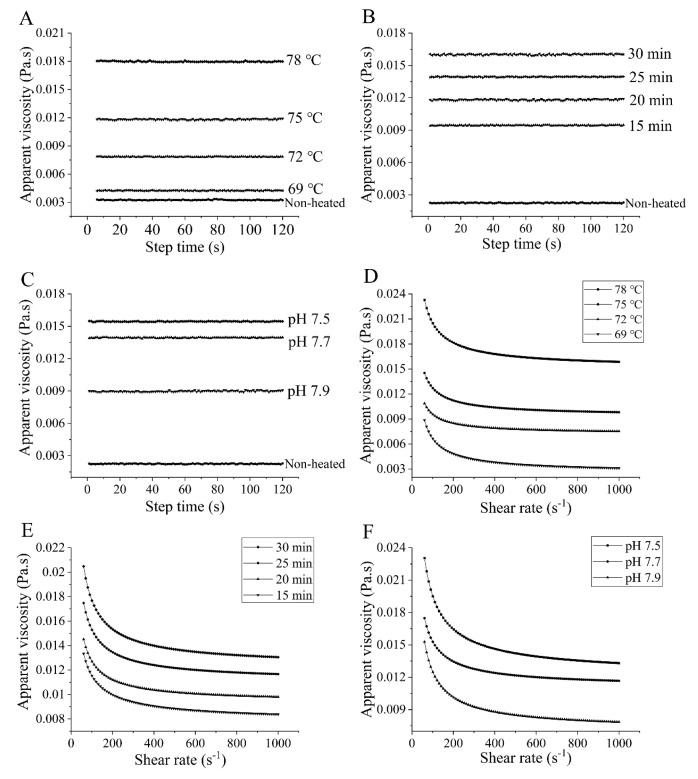
Heat−induced changes in apparent viscosity of goat milk whey protein as a function of shearing time and shear rate, respectively. ((**A**,**D**) heating temperature, (**B**,**E**) heating time, (**C**,**F**) pH). Different letters are significantly different (*p <* 0.05). Error bars represent the standard deviation of the means.

**Figure 5 foods-12-03626-f005:**
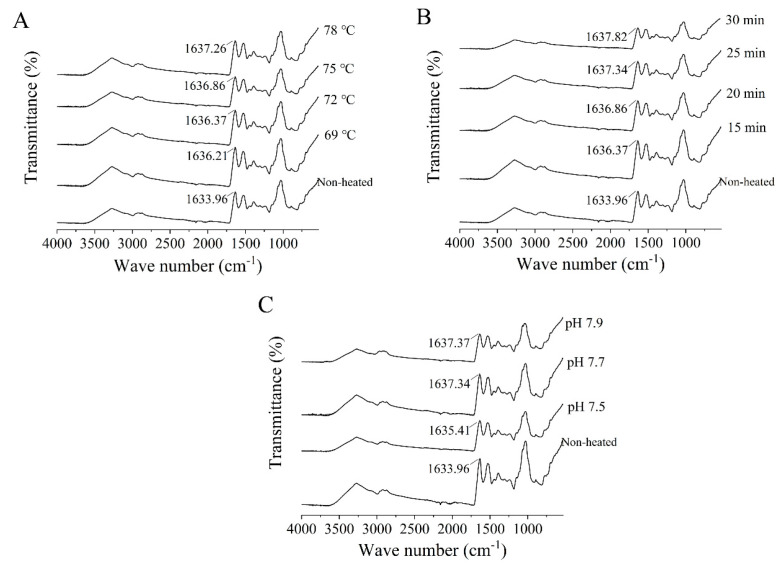
Heat−induced changes in the Fourier−transform infrared spectra of goat milk whey protein ((**A**) heating temperature, (**B**) heating time, (**C**) pH).

**Figure 6 foods-12-03626-f006:**
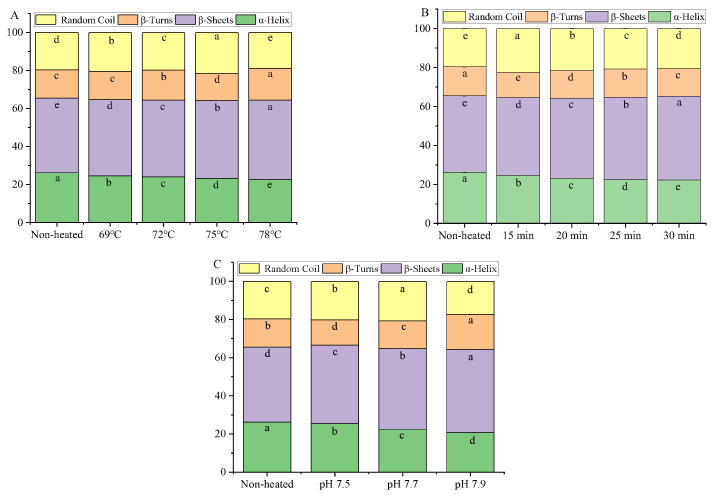
Heat−induced changes in the secondary structure contents of goat milk whey protein ((**A**) heating temperature, (**B**) heating time, (**C**) pH). Different letters are significantly different (*p <* 0.05). Error bars represent the standard deviation of the means.

**Figure 7 foods-12-03626-f007:**
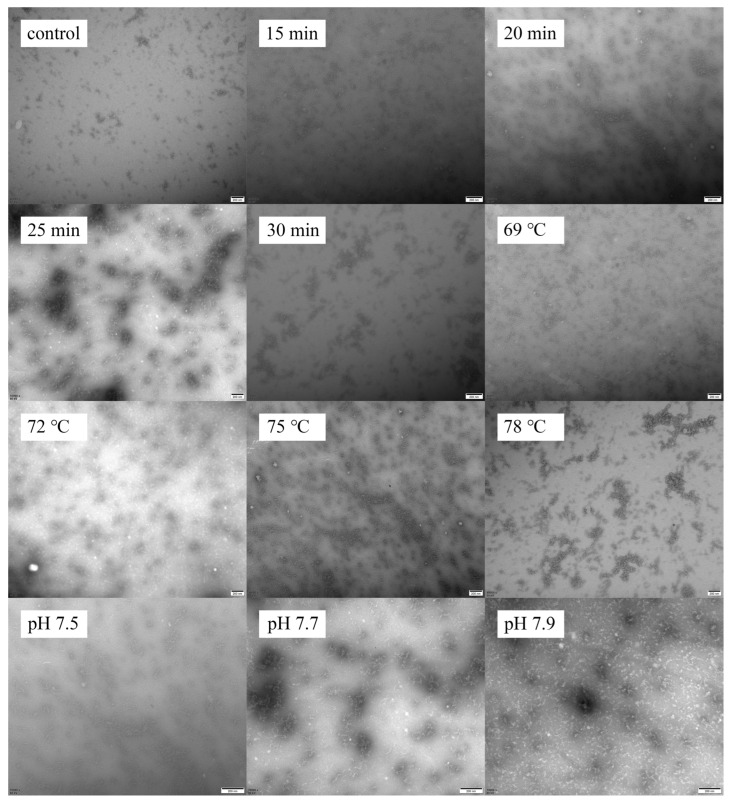
Heat−induced changes in microscopy images of goat milk whey protein.

**Figure 8 foods-12-03626-f008:**
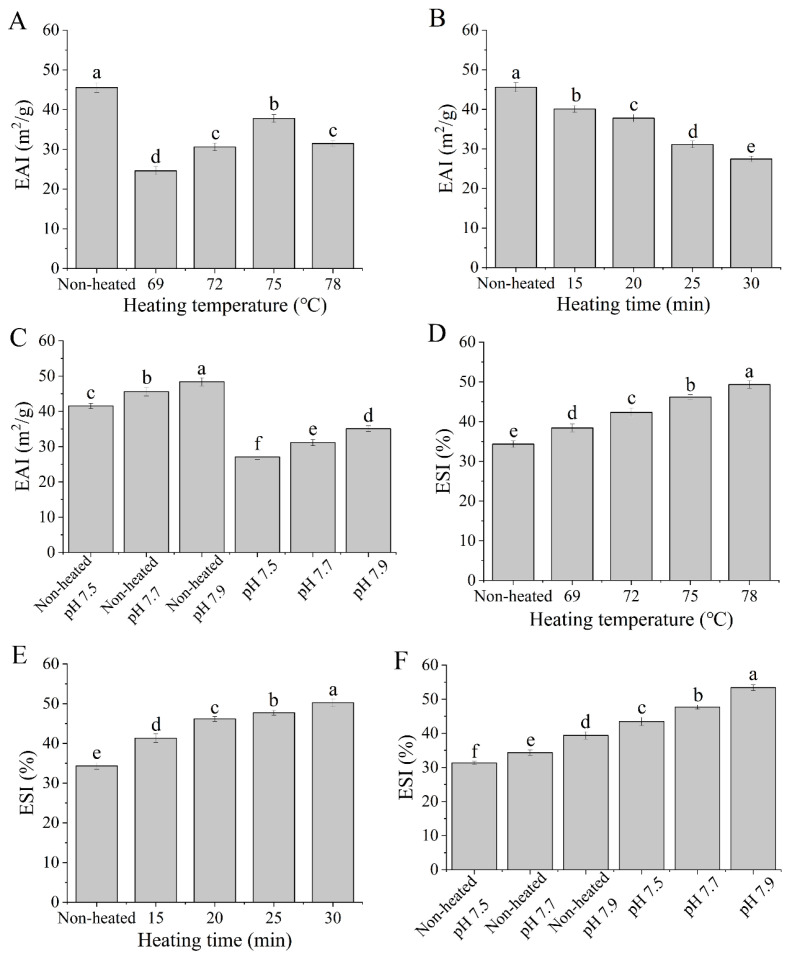
Heat−induced changes in the emulsifying properties of goat milk whey protein ((**A**,**D**) heating temperature, (**B**,**E**) heating time, (**C**,**F**) pH). EAI = emulsifying activity index; ESI = emulsion stability index. Different letters are significantly different (*p <* 0.05). Error bars represent the standard deviation of the means.

**Figure 9 foods-12-03626-f009:**
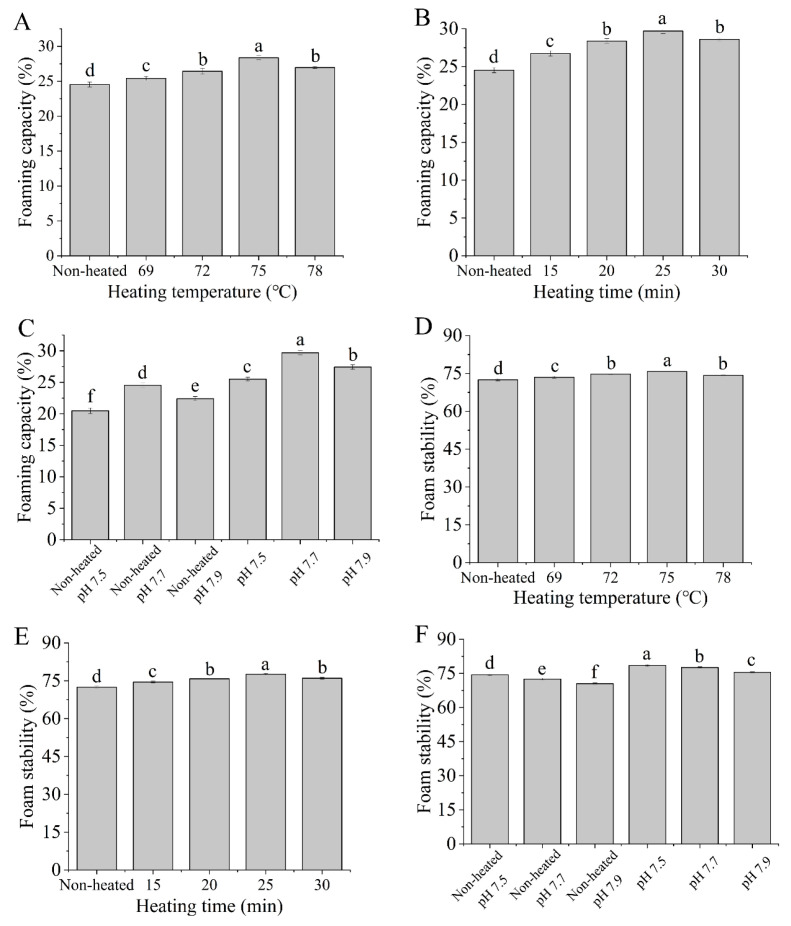
Heat−induced changes in the foaming capacity and foam stability of goat milk whey protein ((**A**,**D**) heating temperature, (**B**,**E**) heating time, (**C**,**F**) pH). Different letters are significantly different (*p <* 0.05). Error bars represent the standard deviation of the means.

## Data Availability

Data are contained within the article.

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
