# Peer review of "Physicochemical and Functional Properties of Thermal-Induced Polymerized Goat Milk Whey Protein"

_foods, 2023, doi:10.3390/foods12193626_

Round 1
Reviewer 1 Report
The topic discussed in the article is interesting. It might contribute to subproduct valorization and/or as raw element of different biocomposites. Considering these aspects, the authors may consider the following recommendation:
· At the end of the Introduction part, is mentioned the fact that the investigation made on the physicochemical and functional properties of thermal-induced polymerized goat milk whey protein could give information that may be further used. Could the researchers indicate some possible utilisations of the polymerized proteins they studied? It is desirable to use the data presented in the literature and sustain the added value potential brought by the products obtained after utilizing the different regimens studied.
The paper could be perfect after major changes. It has to be revised by the authors and resubmitted with suggested modifications specified in the reviewer’s comments.
Minor editing of English language required.
Author Response
Authors: Thank you for your suggestion. We added some possible utilizations of the polymerized proteins in Lines 48-56 of the revised manuscripts.
Lines 48-56: Our previous research had demonstrated the potential of polymerized goat milk whey protein (PGWP) as an effective thickening agent, enhancing the viscosity and minimizing syneresis in yogurt [10]. Furthermore, PGWP showed promise as an ideal carrier material for delivering bioactive substances such as soy isoflavones [11], thereby extending its utility in functional food applications. Additionally, controlled thermally denaturation of whey protein can serve as an efficient fat substitute and stabilizer in yogurt formulations [12]. It is worth noting that Caner's work revealed that whey protein isolate coatings can act as a protective barrier, prolonging the shelf life of eggs [13].
References:
[10] Tian, M.; Cheng, J.J.; Wang, H.; Xie, Q.G.; Wei, Q.S.; Guo, M.R. Effects of polymerized goat milk whey protein on physicochemical properties and microstructure of recombined goat milk yogurt. J. Dairy Sci. 2021, 105, 21581
[11] Tian, M.; Wang, C.N.; Cheng, J.J.; Wang, H.; Guo, M.R. Preparation and Characterization of Soy Isoflavones Nanoparticles Using Polymerized Goat Milk Whey Protein as Wall Material. Foods. 2020, 9: 9091198.
[12] Li, H.J.; Liu, T.T.; Zou, X.; Yang, C.; Li, H.B.; Cui, W.M.; Yu, J.H. Utilization of thermal-denatured whey protein isolate-milk fat emulsion gel microparticles as stabilizers and fat replacers in low-fat yogurt. LWT-Food Sci. Technol. 2021, 150, 112045.
[13] Caner, C. Whey protein isolate coating and concentration effects on egg shelf life. J. Sci. Food Agric. 2005, 85(13), 2143-2148.
Reviewer 2 Report
Review: Physicochemical and functional properties of thermal induced polymerized goat milk whey protein
In their manuscript, the authors investigate goat-milk whey protein solutions under the impact of different temperature and pH-regimes as well as different treatment times at a given temperature. The authors use different techniques such as dynamic-light scattering or the measurement of the zeta-potential, FTIR-spectroscopy, TEM, and further protein-biochemistry assays. The manuscript is very well written without any flaws.
Remarks:
Line 61: Please mention the region of origin of the milk.
Line 185: Various authors describe conformational changes due to a exposure to lower/higher pHs [1].
Chapter 3.4.: I think you can formulate your conclusion less defensive (temperature regimes). Since the viscosity has increased, it is reasonable to assume that, with no further uptake of other substances, aggregation of the proteins has occurred.
Chapter 3.5.: Less defensive too! It is well known that unfolding proteins make such a shift [2].
- Mills, O.E. and L.K. Creamer, A conformational change in bovine β-lactoglobulin at low pH. Biochimica et Biophysica Acta (BBA) - Protein Structure, 1975. 379(2): p. 618-626.
- Royer, C.A., Fluorescence Spectroscopy, in Protein Stability and Folding: Theory and Practice, B.A. Shirley, Editor. 1995, Humana Press: Totowa, NJ. p. 65-89.
The manuscript is very well written without any flaws.
Author Response
Q 1: Line 61: Please mention the region of origin of the milk.
Authors: Thank you for your suggestion. The milk used in this study was obtained from a local farm (Feihe Dairy Industry Co. Ltd., Harbin, China). The region of origin of the milk was added, please see the revised manuscript in Lines 69-70.
Lines 69-70: Raw goat milk (≥8.15% nonfat solids, 3.86% protein, and 4.02% fat, w/v) was purchased from a local farm (Feihe Dairy Industry Co. Ltd., Harbin, China).
Q 2: Line 185: Various authors describe conformational changes due to an exposure to lower/higher pHs [1].
[1]. Mills, O.E. and L.K. Creamer, A conformational change in bovine β-lactoglobulin at low pH. Biochimica et Biophysica Acta (BBA) - Protein Structure, 1975. 379(2): p. 618-626.
Authors: Thank you for your comment. We agreed. The zeta potential of a system can be influenced by various factors, including pH, electrical conductivity, and the addition of polymers, among others. The change in pH leads to the different conformations in the goat milk whey proteins. Previous research had also illustrated that changes in pH prior to heating can impact the formation and composition of protein aggregates in goat milk during heat treatment (Nair et al., 2013). We have made the necessary modifications to this section in accordance with your suggestions. Please refer to the revised manuscript in Lines 221-227.
Lines 221-227: The observed increase in absolute zeta potential can be attributed to the exposure of buried charged residues due to protein denaturation during heating [28]. However, variations in heating temperature and time did not yield a significant effect on the zeta potential of PGWP samples (P > 0.05). This suggested that the changes in whey protein conformation induced by temperature and time may not be substantial enough to result in significant alterations in the zeta potential of the systems.
References:
Nai P.; Dalgleis D.G.; Corredig M. Colloidal properties of concentrated heated milk. Soft Matter. 2013, 9, 3815-3824.
[28] Zhao Z.; Xiao Q. Effect of chitosan on the heat stability of whey protein solution as a function of pH. J Sci Food Agr. 2017, 97, 1576-1581.
Q 3: 3.4.: I think you can formulate your conclusion less defensive (temperature regimes). Since the viscosity has increased, it is reasonable to assume that, with no further uptake of other substances, aggregation of the proteins has occurred.
Authors: We agreed with your comment and this section had been revised. Please see the revised manuscript in Lines 278-285.
Lines 278-285: The process of heating leads to the disruption of various intramolecular bonds that stabilize the native goat milk whey protein structure. Beyond a certain temperature, protein molecules unfold and subsequently aggregate. These aggregates tend to be larger in size, exhibit greater asymmetry in shape, and possess a larger effective volume fraction compared to the native molecules, resulting in an increase in viscosity. However, it's essential to note that viscosity is influenced by numerous molecular properties, including size, shape, and flexibility [35]. All of these factors may be contributed to the observed increase in viscosity.
Reference:
[35] Vardhanabhuti, B.; Foegeding, E.A. Rheological properties and characterization of polymerized whey protein isolates. J Agr Food Chem. 1999, 47, 36-49.
Q 4: Chapter 3.5.: Less defensive too! It is well known that unfolding proteins make such a shift [2].
[2]. Royer, C.A., Fluorescence Spectroscopy, in Protein Stability and Folding: Theory and Practice, B.A. Shirley, Editor. 1995, Humana Press: Totowa, NJ. p. 65-89.
Authors: Thank you for your comment. This section has been revised according to your suggestion. Please see the revised manuscript in Lines 303-311.
Lines 303-311: The amide I band, which falls within the spectral region between 1600 and 1700 cm-1, is sensitive to alterations in the secondary structure of proteins. It represents the stretching vibrations of C=O bonds and is influenced by hydrogen bonding interactions [38]. In the FTIR spectra of the unheated samples, a distinctive peak at 1633.96 cm-1 was prominently observed. Following heat treatment, significant blueshifts were observed as the heating temperature and time increased. These alterations in heat treatment conditions influenced the conformation of proteins by disrupting intramolecular hydrogen bonds, leading to a shift of the spectral peaks towards higher wavenumbers [39].
Round 2
Reviewer 1 Report
The authors took into consideration the recommendation made.
Minor editing of English language required.